# Graphene Oxide and Graphene Reinforced PMMA Bone Cements: Evaluation of Thermal Properties and Biocompatibility

**DOI:** 10.3390/ma12193146

**Published:** 2019-09-26

**Authors:** E. Paz, Y. Ballesteros, J. Abenojar, J.C. del Real, N.J. Dunne

**Affiliations:** 1Institute for Research in Technology /Mechanical Engineering Dept., Universidad Pontificia Comillas, Alberto Aguilera 25, 28015 Madrid, Spain; yballesteros@comillas.edu (Y.B.);; 2Materials Science and Engineering Department, IAAB, Materials Performance Group, Universidad Carlos III de Madrid, Av. Universidad 30, 28911 Leganes, Madrid, Spain; abenojar@ing.uc3m.es; 3Centre for Medical Engineering Research, School of Mechanical and Manufacturing Engineering, Dublin City University, Stokes Building, Collins Avenue, Dublin 9, Ireland; 4School of Mechanical and Manufacturing Engineering, Dublin City University, Stokes Building, Collins Avenue, Dublin 9, Ireland; 5School of Pharmacy, Queen’s University Belfast, 97 Lisburn Road, Belfast BT9 7BL, UK; 6Trinity Centre for Bioengineering, Trinity Biomedical Sciences Institute, Trinity College Dublin, Dublin 2, Ireland; 7Advanced Materials and Bioengineering Research Centre (AMBER), Royal College of Surgeons in Ireland and Trinity College Dublin, Dublin D02 YN77, Ireland

**Keywords:** bone cement, graphene oxide, thermal properties, thermal conductivity, kinetics, biocompatibility

## Abstract

The incorporation of well-dispersed graphene oxide (GO) and graphene (G) has been demonstrated as a promising solution to improve the mechanical performance of polymethyl methacrylate (PMMA) bone cements in an attempt to enhance the long-term survival of the cemented orthopaedic implants. However, to move forward with the clinical application of graphene-based PMMA bone cements, it is necessary to ensure the incorporation of graphene-based powders do not negatively affect other fundamental properties (e.g., thermal properties and biocompatibility), which may compromise the clinical success of the implant. In this study, the effect of incorporating GO and G on thermal properties, biocompatibility, and antimicrobial activity of PMMA bone cement was investigated. Differential scanning calorimetry studies demonstrated that the extent of the polymerisation reaction, heat generation, thermal conductivity, or glass transition temperature were not significantly (*p* > 0.05) affected by the addition of the GO or G powders. The cell viability showed no significant difference (*p* > 0.05) in viability when MC3-T3 cells were exposed to the surface of G- or GO-PMMA bone cements in comparison to the control. In conclusion, this study demonstrated the incorporation of GO or G powder did not significantly influence the thermal properties or biocompatibility of PMMA bone cements, potentially allowing its clinical progression.

## 1. Introduction

Graphene oxide (GO) and graphene (G) have become two of the most interesting nanomaterials due to their unique properties (i.e., high mechanical strength, high specific area, easy functionalisation, high thermal and electrical conductivity, catalyst capacity, etc.) [1,2,3,4,5,6]. These unique properties could be of great value for creating advanced biomaterials (i.e., reinforcement capability, antimicrobial activity, electrical conductivity, drug carrier capability, etc.) [7,8].

The potential for carbon-based nanomaterials such as nanofillers to enhance the mechanical properties of polymeric materials has increased in recent years. Significant improvements in fatigue life and fracture toughness of polymer-based nanocomposites (e.g., epoxy, acrylic, polyurethane, etc.) can be achieved via the incorporation of carbon-based nanofillers. It has been postulated that a good dispersion of these nanofillers within the polymer matrix produces a deviation and detention of crack fronts during their propagation, increasing the required energy for failure. These nanofillers enhance the toughness at very low loading levels due to their high surface area, which promotes a strong interaction between the nanofiller and the polymer-based matrix. It has been reported that a more homogenous dispersion of the nanofiller and stronger interfacial bond between the filler and matrix will result in a greater improvement in the mechanical performance [9].

The use of GO as a reinforcing agent has demonstrated particular advantages over other carbon-based nanomaterials since it is readily obtainable in large quantities, relatively easy to exfoliate and homogenously disperse, and has functional groups available to facilitate strong interfacial bonding with the polymeric matrix [4,10].

Recently, the development of advanced bone cements for orthopaedic applications using GO and G has been investigated [11,12]. The incorporation of well-dispersed GO and G powders have been demonstrated as promising solutions to improve the mechanical performance of polymethyl methacrylate (PMMA) bone cements in an attempt to enhance the long-term survival of the cemented orthopaedic implants. 

In a previous study, the effect of loading different levels of GO and G powder on the mechanical properties of PMMA bone cements was investigated [11]. In these studies, improvements in the mechanical performance under static and fatigue conditions was achieved with the addition of 0.1 wt.% GO or G powder. Additionally, it was observed that the incorporation of GO powder provided better results when compared to G powder due to the presence of functional groups on the surface of the GO powder, which improved the dispersion and facilitated strong interfacial bonds between the GO and the PMMA matrix.

Despite these promising results, to move forward with the clinical translation of GO- and G-PMMA bone cements, it is necessary to ensure the incorporation of a graphene-based powder does not negatively affect the thermal properties and biocompatibility of the bone cement, which could compromise the success of the implant.

Prior to their clinical application, PMMA-bone cements are formed by two phases: liquid and powder. The powder phase is mainly composed of the pre-polymerised PMMA, a radiopaque agent (barium sulphate or zirconium oxide), and an activator of the polymerisation reaction (benzoyl peroxide). The liquid phase is composed by the monomer (methyl methacrylate) and the initiator of the reaction (dimethyl-p-toluidine). During joint replacement surgery, the powder and liquid phases are mixed together under vacuum to form a viscous mass, which is easily delivered into the intramedullary canal prior to placement of the orthopaedic implant. The polymerisation reaction of the bone cement begins when the benzoyl peroxide from the powder phase comes into contact with the dimethyl-p-toluidine from the liquid phase. This reaction produces free radical species that initiate the polymerisation of methyl methacrylate (MMA) monomer molecules. It is important to take into account that this free-radical polymerisation, which leads to the hardening of the PMMA bone cement, is relatively fast (<15–30 min) and highly exothermic. During polymerisation, the exothermic temperature can reach 80 °C in the cement mantle [13,14,15,16].

A good understanding of the reaction parameters is fundamental to ensure the feasibility of the new bone cement formulations. An acceleration in the polymerisation reaction would be detrimental because it would increase the polymerisation temperature and result in the thermal necrosis of the surrounding tissue. Conversely, retardation in the polymerisation reaction would extend the setting time of the PMMA bone cement, which would impair alignment of the implant and increase the negative effect of the residual monomer.

It has been reported that the addition of certain adjuvants to the PMMA-based bone cement can produce important variations in the polymerisation reaction. For example, it has been found that the addition of multiwalled carbon nanotubes (MWCNT) to PMMA bone cement retards the polymerisation reaction because MWCNT powder acts as a free radical scavenger and inhibits the polymerisation reaction [17,18,19]. A decrease in the maximum polymerisation temperature and a delay in the setting time have been reported when 0.1–1 wt.% of MWCNT powder was incorporated into PMMA bone cement [17]. 

An adequate level of biocompatibility is a critical parameter when developing new bone cement formulations [8,20,21,22]. It has been reported that the toxicity of carbon-based nanomaterials strongly depends on several factors, for example, the synthesis route, concentration, time, and exposure [23,24]. Studies have shown the toxic effect of carbon-based nanomaterials decreases when they are used as material reinforcement encapsulated within a polymer-based matrix [8]. Moreover, different studies have reported that GO and G reinforced biomaterials have demonstrated adequate biocompatibility, and in some cases, the addition of GO powder provided increased antimicrobial activity [7,25,26,27].

The aim of this study was to investigate the effect of the GO or G powder incorporation on the thermal properties, biocompatibility, and antimicrobial activity of PMMA bone cement. Specifically, 0.1 wt.% of GO or G powder was incorporated into a normal viscosity bone cement, and the heat generated during polymerisation, degree of conversion (i.e., residual monomer), kinetic parameters, glass transition temperature, and thermal conductivity were determined using differential scanning calorimetry (DSC). Additionally, the cell viability of MC3-T3 cells following 72 h of direct contact with the GO- and G-PMMA bone cements and antimicrobial activity against *Staphylococcus aureus* were investigated.

## 2. Materials and Method

### 2.1. Materials

#### 2.1.1. Nanomaterials

Powders of graphene oxide (GO) (NanoInnova Technologies, Spain) or graphene (G) (Avanzare Nanotechnology, Spain) were incorporated into the PMMA bone cement. According the supplier data sheets, the GO sheets had an average lateral size of 1.8–2.7 nm and a thickness of 0.7–1.2 nm and the G powder was composed of 1–2 layers of graphene sheets with an average lateral size of 50–500 nm and a thickness of 0.7 nm.

#### 2.1.2. PMMA Bone Cement

Similar to other PMMA bone cements, the PMMA bone cement used in this study was a two-phase bone cement. The solid phase, composed of 36.36 g of Colacryl B866 (Lucite International Ltd., Billingham, UK), was a pre-polymerised PMMA powder that was supplied pre-blended with the polymerisation initiator (benzoyl peroxide). Additionally, 3.64 g of barium sulphate (Sigma Aldrich, Madrid, Spain) was incorporated into the powder phase. The liquid phase was composed of 19.9 mL of the methyl methacrylate (MMA) monomer and 160 µL of an activator (N,N-Dimethyl-p-toluidine), which were supplied by Sigma Aldrich (Madrid, Spain). The bone cement formulation described has been used in previous studies and is analogous to the commercial bone cement, DePuy CMW 1 [11,28].

Then, 0.1 wt.% of GO or G powder was incorporated into the PMMA bone cement as previous studies demonstrated it is the optimal loading level in terms of improving the mechanical performance [11]. Prior to mixing the powder and liquid phases, 40 mg of GO or G powder was homogenously dispersed in the liquid phase using ultrasonication with a Digital Sonifier 450 (Branson Ultrasonics Corporation, Danbury, CT, USA). Subsequently, the liquid phase was sonicated at 50% amplitude for 3 min. To prevent overheating during ultrasonication, the vial with the liquid phase was held at 22 ± 1 °C in a water bath. Following ultrasonication, to reduce bubble formation, the suspension was placed in an ultrasonic bath (Elmasonic p60h, Elma Schmidbauer GmbH, Singen Germany) for 1 min. PMMA bone cement without G or GO powder was used as the control cement for comparative purposes.

Bone cement samples were prepared as per the manufacturer’s instructions; the polymer and liquid phases were mixed using a HiVac^®^ Vacuum Mixing System (Summit Medical, Cheltenham, UK) under a reduced pressure of 70.0 ± 0.1 kPa. The bone cement was prepared under ambient conditions (22 ± 1 °C) and at a relative humidity of not less than 40%.

### 2.2. Experimental Procedure

#### 2.2.1. Differential Scanning Calorimetry (DSC)

Differential scanning calorimetry (DSC) was used to conduct thermal analysis. Specifically, a Mettler Toledo DSC822 (Madrid, Spain) was used to determine the main characteristics during the polymerisation reaction, in other words, the heat generated during polymerisation, residual monomers, and kinetic parameters. Additionally, the thermal properties of the polymerised bone cement were determined, in other words, the glass transition temperature (*T_g_*) and thermal conductivity. All DSC tests were conducted using an aluminium crucible with a capacity of 40 µL and a 50 m hole in the lid; the amount of sample tested was between 10–15 mg. Nitrogen was used as the purge gas and was delivered at a rate of 80 mL/min.

##### Kinetic Analysis of Polymerisation Reaction

To determine the influence of GO and G powder on the polymerisation reaction process for the PMMA bone cement, non-isothermal tests were performed from 0 to 200 °C at three different scan rates: 5, 10 and 20 °C/min. Each bone cement sample was placed in the aluminium pan at 3 min from the start of cement mixing. The DSC thermogram of heat flow versus polymerisation time was obtained for each test.

The heat released during the polymerisation (Δ*H*), expressed in J/mol, was determined as the area under the heat flow versus polymerisation time curve.

The conversion degree reached at a certain time (*α_t_*) was calculated using Equation (1), where Δ*H_t_* is the reaction enthalpy at time *t* (area under the curve at time t) and Δ*H_P_* is the total reaction enthalpy (total area under the curve).

(1)$$\alpha =\;\frac{\Delta H_{t}}{\Delta H_{P}}$$

From the Equation (1), the general equation to calculate the reaction rate can be used (Equation (2)), where *k* is the reaction rate constant, and *f(x)* is a function that depends on concentration and time.

(2)$$kf\left(x\right)=\;\frac{d\alpha }{dt}$$

Each obtained thermogram was analysed using the STARe Software (Mettler Toledo, Madrid, Spain) based on the Model Free Kinetic analysis (MFK) [29,30]. This model provides the activation energy as a function of conversion degree and allows the simulation of a model that predicts the conversion degree in an isothermal process using the data obtained from non-isothermal tests.

To compare the isothermal degree conversion curves obtained from the MFK simulation, an isothermal test was performed at 25 °C for 60 min. A second segment of dynamic scanning from 25 to 200 °C at a scan rate of 10 °C/min was included with the isothermal scan to complete the polymerisation reaction and determine the free residual monomer content following isothermal polymerisation of the PMMA bone cement. Three samples were tested for each cement type.

##### Glass Transition Temperature

To determine the *T_g_,* a sample of polymerised bone cement was tested using DSC from 0 to 200 °C at 5 °C/min. Two passes were performed on each sample: the first to eliminate any absorbed moisture and the unpolymerised monomer, and the second to measure the *T_g_*. The drop in the thermogram line provides information on the change in the heat capacity of the bone cement when heated above its *T_g_.* Three samples were tested for each cement type.

##### Thermal Conductivity

The thermal conductivity of the polymerised bone cement was determined using DSC following the method of Hakvoort and van Reijen [31,32]. Specifically, each polymerised bone cement sample was placed in the sample furnace in the sample position. An aluminium crucible with a calibration substance (gallium) was then placed on top of the cement sample. An empty aluminium crucible was placed in the reference position of the furnace.

Each sample was a circular disk of polymerised bone cement with a height of 1.5 ± 0.1 mm and 6 ± 0.1 mm of diameter. To ensure good heat transfer, oil was applied on both faces of the sample. A thermogram was recorded during the melting of the gallium reference. The temperature program used was 28–38 °C at 0.5 °C/min, followed by cooling to −5 °C at 10 °C/min to ensure that the gallium solidified following measurement. Nitrogen was used as the purge gas at a rate of 50 mL/min to prevent oxidation of the gallium. The thermogram of the crucible with the gallium without the cement was used as the reference. The thermal conductivity was calculated using Equation (3), where Δ*h* is the height of the cement sample and *A* is the surface area of the disk samples. *S* is the slope of the linear side of the melting peak obtained from the thermogram (heat flow vs. temperature), *S_1_* is the slope of the thermogram of the sample, and *S_2_* the slope of the thermogram for the crucible with gallium. At least 10 samples of each cement type were measured.

(3)$$\lambda =\frac{∆h}{A\left(\frac{1}{S_{2}}-\frac{1}{S_{1}}\right)}$$

#### 2.2.2. Biocompatibility Tests and Antimicrobial Activity

To study the extent of biocompatibility, disk samples of bone cement (thickness of 2.0 ± 0.1 mm and diameter of 12 ± 0.1 mm) were incubated with an osteoblast precursor cell line (MC3-T3) derived from mouse calvaria. The level of MC3T3 viability was determined after 72 h in culture medium using the CellTiter 96 aqueous cell proliferation assay (Promega, Madison, WI, USA). Following an incubation period of 72 h, 20 µL of MTS reagent was added to each sample with 100 µL of culture medium. The sample was then incubated for 4 h at 37 °C in a humidified 5% CO_2_ atmosphere. The absorbance was recorded at 490 nm using a Universal Microplate Reader EL 800 (BioTek Instruments, Inc., Winooski, VT, USA). The absorbance values recorded were determined to be proportional to the number of viable cells proliferating on each cement surface. The correlation between the number of cells and the absorbance at 490 nm was estimated and was used to calculate the number of viable cells in each sample. Six samples were tested for each cement type.

The antimicrobial activity of the different bone cement samples was assessed using the disk diffusion test. The entire area of the Agar plates was homogeneously seeded with *Staphylococcus aureus* (ATCC^®^29213TM) by inoculation with an overnight broth culture adjusted to MacFarland 0.5 turbidity. Bone cement samples (thickness of 2.0 ± 0.1 mm and diameter of 12 ± 0.1 mm) were placed in the centre of the Agar plate. The zone of inhibition was measured following an incubation period of 24 h, and samples were photographed. Three samples were tested for each cement type.

#### 2.2.3. Statistical Analysis

Each property was expressed as mean ± standard deviation. The results were also evaluated for statistical significance using a one-way analysis of variance (ANOVA) test with a post-hoc Scheffe’s test (SPSS 15.0 for Windows; IBM SPSS, Armonk, NY, USA). A *p*-value less than 0.05 was indicative of statistical significance.

## 3. Results and Discussion

### 3.1. Kinetic of Polymerisation Reaction

The DSC thermograms for each of the cement types tested exhibited two exothermic peaks during the heating of the unpolymerised cement samples (Figure 1). Depending on the heating rate, the first peak appeared at a temperature between 50 and 70 °C. This peak was attributed to the amount of heat released during the non-isothermal curing of the cement (Δ*H_c_*). The second peak appeared between 137 and 151 °C. This peak was notably lower and it was attributed to the residual monomer (Δ*H_R_*) as a consequence of incomplete polymerisation [33,34].

The enthalpy associated with each peak for the different cement types (control, 0.1 wt.% GO- and 0.1 wt.% G-PMMA bone cements) at heating rates of 5, 10 and 20 °C/min is shown in Table 1. For all cement types tested, the values recorded for the polymerisation heat released during the non-isothermal polymerisation (Δ*H_c_*) and the residual monomer (Δ*H_R_*) were similar irrespective of the heating rate. It was also observed that for all cement types, the heat associated with the residual monomer was ~6% of the total released heat (Δ*H_c_* + Δ*H_R_*), which means that independent of the heating rate, the maximum degree of conversion achieved was ~94%.

The data recorded for the GO- and G-PMMA bone cements were similar to the control cement, which indicated the incorporation of GO or G powder at 0.1 wt.% did not produce significant variations in the polymerisation heat or residual monomer content when tested under non-isothermal conditions. The minor differences noted are within the expected error associated with the DSC measurement, sample preparation, and data analysis.

The heat released during polymerisation mainly depends on two aspects: (1) the extent of the polymerisation reaction and (2) the mechanism by which the reaction occurs. This means that for two systems containing identical reactant composition, the released heat during polymerisation is proportional to the extent of the reaction. Conversely, for two systems containing different reactant compositions, the amount of heat released may be different even if the extent of reactions for both systems are identical [35]. Considering this study, the extent of the reaction was not affected by the heating rate or by the incorporation of GO or G powder, which means incompletion of the polymerisation reaction for the PMMA bone cement may be limited by other factors, for example, insufficient activator agent, ratio of activator to initiator, or level of monomer [36]. No clear trends have been reported in the literature regarding the effect of the heating rate or material composition on the heat released during the non-isothermal polymerisation of the PMMA bone cements [34,37].

To determine the effect of GO and G powder on the polymerisation reaction of the PMMA bone cement, the conversion degree as a function of the temperature was calculated using the DSC thermograms for each heating rate (Figure 2). It was taken into account that the maximum conversion degree obtained was ~94%.

The conversion degree curves for the GO-PMMA bone cement at each heating rate are shown in Figure 2A. It can be observed that at higher heating rates, a higher temperature was required to achieve a certain conversion; a similar trend was noted for the control and G-PMMA bone cement. It is suggested that when the heating rate is low, the system remains longer within a temperature range where the reaction can take place, which results in a higher reaction extent [35].

When the conversion degree of GO- and G-PMMA bone cements were compared with the control (Figure 2B), a slight increase was observed in the conversion rate at the beginning of the reaction. For example, at 10 °C/min and 60 °C, the conversion for the control cement was ~35%, and in the case of GO- and G-PMMA bone cements, it was 83% and 60.3%, respectively. However, this initial acceleration in the reaction was largely irrelevant as the final extent of reaction and the start and end reaction temperature did not change significantly.

The activation energy (E_a_) as a function of the conversion degree (Figure 3) was obtained from the conversion curves (Figure 2) at the three different heating rates using the MFK STARe Software (Mettler Toledo, Madrid, Spain). For the control bone cement, the highest values of *E_a_* were obtained at the beginning of the reaction and then the *E_a_* decreased continually. This decrease was especially pronounced during the last stages of the reaction (approximately after 75% of conversion).

This trend can be explained if the three phases of the free radical polymerisation reaction are considered: (1) initiation, (2) propagation, and (3) termination [33,35,38,39]. During the initiation phase, the formation of the free radical species requires a higher energy consumption and consequently the *E_a_* during this step is high. However, once the reaction is initiated, the energy required during the second phase (propagation) is relatively low. During the propagation phase, it is common for the reaction to undergo auto-acceleration (also known as gel effect) [40]. This results in an increase in the conversion rate during the intermediate phase of the free radical polymerisation as a consequence of an increase in the viscosity. The high viscosity produces a decrease in chain mobility, which hampers the termination of the polymeric chains and produces a rapid increase in the overall rate of reaction. A decrease in the activation energy in observed during auto-acceleration. Termination becomes more difficult as the growing chains are unable to diffuse, making it difficult for the two reactive species to meet. This is the reason that PMMA-based bone cements achieve a conversion degree of 94% [41].

However, in comparison with the control bone cement, the addition of GO and G powder produced notable changes in the development of the polymerisation reaction (Figure 3). One of the most notable differences was that the E_a_ required at the start of the polymerisation reaction for the GO- and G-PMMA bone cements was lower than that for the control. The *E_a_* remained lower for the GO- and G-PMMA bone cements during almost all of the reaction until a conversion degree of ~75%. This observation suggests the presence of G and GO powder can act as a catalyst, thereby favouring reaction initiation and the initial steps of the propagation. The catalytic activity of GO and G powder has been previously reported [1,42,43], in particular, their ability to accelerate the free radical reactions [44]. This is attributed to the acidity of GO and G powders and their oxidising properties, which depends on the functional groups (e.g., hydroxyl or carboxyl groups) present on the surface [42]. The higher levels of functionalisation of the GO can explain the higher acceleration effect observed for the GO-PMMA bone cements.

Once the conversion degree was reached, the catalytic action of GO and G powders ceased, then the *E_a_* for the GO- and G-PMMA bone cements increased. This is because an extra supply of energy was required to continue with the conversion in order to complete the polymerisation reaction; this increase was particularly evident when the GO powder was incorporated into the PMMA bone cement. This trend suggests the GO and G powders, which initially favoured radical initiation, could in turn have consumed some of the active centres, making it more difficult for chain termination to occur. This also can be attributed to the lower diffusion capacity for the methyl methacrylate monomer through the polymerising mass due to the presence of the GO powder, which can make polymer chain and radical mobility more difficult [42]. This mechanism has been observed during the polymerisation reaction of different polymers in the presence of nano-sized powders [45,46,47].

A simulation of the conversion degree versus the polymerisation reaction time as a function of the temperature was obtained using MFK analysis (Table 2). For each bone cement type, the time required to complete the cement polymerisation was lower as the temperature increased.

To corroborate the MFK simulation data, the isothermal conversion degree obtained by simulation at 25 °C was compared with the DSC isothermal data at the same temperature. For the G-PMMA bone cement, it was observed that although the DSC isothermal data showed complete polymerisation at 75 min and the MFK model predicted longer times (~165 min), the trend of the simulated conversion degree curve and the one corresponding to the experimental data are similar (Figure 4). Similar findings were noted for the control and G-PMMA bone cement types.

Comparing the simulated conversion degree data (Table 2), it is noted that for all temperatures, the polymerisation time was lower for the control when compared to the GO- and G-PMMA bone cements. Moreover, incorporation of the GO powder into the PMMA bone cement extended the reaction time when compared to the G powder. For example, at 25 °C, the control cement reached its maximum polymerisation after 21 min, while the GO- and G-PMMA bone cements required 660 min and 165 min, respectively, to achieve the same conversion degree. This difference in conversion degree was only notable at the high conversion degree levels (i.e., ≥80%). However, no notable differences were observed in the time taken to reach a conversion degree of 80%: 21.03 min for control, 25.04 min for G, and 25.49 min for GO. It can be postulated that a PMMA bone cement demonstrating a conversion degree of 80% can exhibit sufficient mechanical performance and adequate fixation; therefore, the incorporation of GO or G powder to PMMA bone cement may not present an issue in terms of implant fixation, patient recovery, and long-term implant stability [48].

It is also interesting to note that for GO powder, the time to progress the polymerisation reaction from 90% to 94% was extremely long (Table 2), in other words, the 90% of conversion was reached in approximately 100 min, but it took 31 h to progress the reaction from 90% to 94% of conversion. This observation supports the fact that the termination reaction in the case of the GO-PMMA bone cement was delayed due the presence of the GO powder. This deceleration in the polymerisation during the latter stages can explain the increase in the residual monomer when GO or G powders were incorporated into PMMA-based bone cements [11,17,49]. Finally, it is important to note the residual monomer levels determined from the non-isothermal and isothermal DSC analysis do not have to be the same. This is because during the non-isothermal DSC analysis, the maximum level of polymerisation is achieved, and in contrast, the degree of polymerisation reached is a function of time for the isothermal DSC tests.

### 3.2. Glass Transition Temperature and Thermal Conductivity

The incorporation of G or GO powder into the PMMA bone cement at the 0.1 wt.% level did not significantly influence (*p* > 0.05) the *T_g_* or thermal conductivity when compared to the control (Table 3).

Other studies have reported a reduction in polymer chain mobility when GO or G powder was incorporated into a PMMA-based bone cement, which resulted in an increase in the *T_g_* [49,50]. However, this increase in *T_g_* was only observed at levels of GO or G powder loading greater than 0.5 wt.% [49,50,51]. DSC analysis from this study indicated a 2% reduction in *T_g_* when 0.1 wt.% G powder was added to the PMMA bone cement. However, this difference was insignificant (*p* < 0.05) and was within the typical standard deviation associated with the DSC technique. Similarly, incorporating 0.1 wt.% GO powder into the PMMA bone cement resulted in a 0.09% increase in T_g_ when compared to the control.

It has been observed that the incorporation of carbon-based nanomaterials significantly increases the thermal conductivity of different polymeric-based materials [4,52,53,54,55]. Considering PMMA-based bone cements, an increase in thermal conductivity could be advantageous in terms of heat dissipation during the polymerisation reaction, potentially reducing the risk of thermal necrosis. Data from this study showed a statistically insignificant (*p* > 0.05) increase in thermal conductivity when G powder was incorporated into the PMMA bone cement. However, no increase in thermal conductivity was observed when the GO powder was incorporated into the PMMA bone cement. These findings are not noteworthy as a higher percolation threshold is required to improve thermal conductivity of polymer-based materials [53].

### 3.3. Cell Viability and Antimicrobial Activity

Although the biocompatibility of GO and G powder when incorporated into biomaterials has been previously reported [20,23,49,56], the evaluation of the cytotoxicity must be conducted to ensure the use of these GO- and G-PMMA bone cements have potential *in vivo*. The viability of the MC3-T3 cells when exposed to the different bone cement types as a function of viable cell number following an incubation period of 72 h are reported in Table 4. The results indicate that the incorporation of GO or G powder to PMMA bone cement did not invoke a cytotoxic response, thereby demonstrating an adequate level of biocompatibility. Additionally, no statistical difference (*p* > 0.05) was observed when the control cement was compared to the GO- or G-PMMA bone cements with respect to viable cell number following an incubation period of 72 h.

Figure 5 shows the extent of antimicrobial activity for the control, GO-, and G-PMMA bone cements. No evidence of an inhibition zone was observed for any of the bone cement types, which means that incorporating 0.1 wt.% of G or GO powder to the PMMA bone cement demonstrated no antimicrobial activity against *Staphylococcus aureus*. Some studies have reported that the presence of GO powder can improve the antimicrobial activity and consequently has been used to develop antibacterial activity in some inert materials [7,23,49]. A study has demonstrated that the antimicrobial activity is dependent on different factors, for example, GO sheet dimension and the loading level of the nano-sized powder [5,24,42,51]. Additionally, these studies investigating the antibacterial activity of GO-based materials contained higher loading levels of GO powder than 0.1 wt.%, which suggests the main reason for lack of antimicrobial activity being reported in this study is because the level of G or GO powder was below the threshold required for a positive antimicrobial response. A level of loading of 0.1 wt.% GO and G powder has been demonstrated as optimal in terms of static and dynamic mechanical properties. However, in light of the results from this study, future works should focus on the effect of G and GO powder loading levels on the thermal properties, antimicrobial activity, and biological response of PMMA bone cement in an effort to establish the optimal loading.

## 4. Conclusions

The incorporation of GO or G powder to PMMA bone cement at a loading level of 0.1 wt.% has potential for use during joint replacement surgery. In this study, the thermal properties or the polymerisation reaction for PMMA bone cement did not demonstrate any significant change on incorporation of GO or G powders. Additionally, the GO- and G-PMMA bone cements exhibited no cytotoxic response and an adequate level of biocompatibility. However, the incorporation of GO or G powder at a 0.1 wt.% level of loading did not demonstrate an improvement in antimicrobial activity or increase of the thermal conductivity of the PMMA bone cement. The latter parameter could be considered important in terms of reducing the extent of thermal necrosis.

## Figures and Tables

**Figure 1 materials-12-03146-f001:**
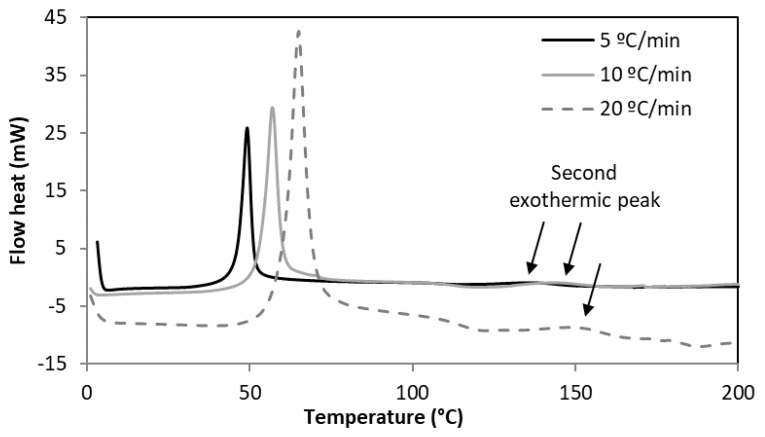
DSC thermogram for the control bone cement samples when subjected to different heating rates.

**Figure 2 materials-12-03146-f002:**
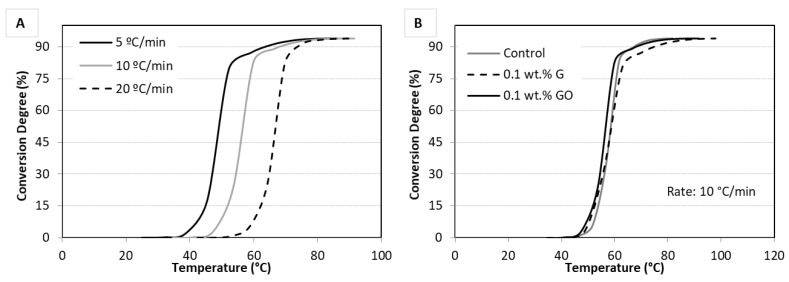
Conversion degree curves for (**A**) polymethyl methacrylate (PMMA) bone cement with 0.1 wt.% GO powder when subjected at the different heating rates and (**B**) the different bone cement types tested at a heating rate of 10 °C/min.

**Figure 3 materials-12-03146-f003:**
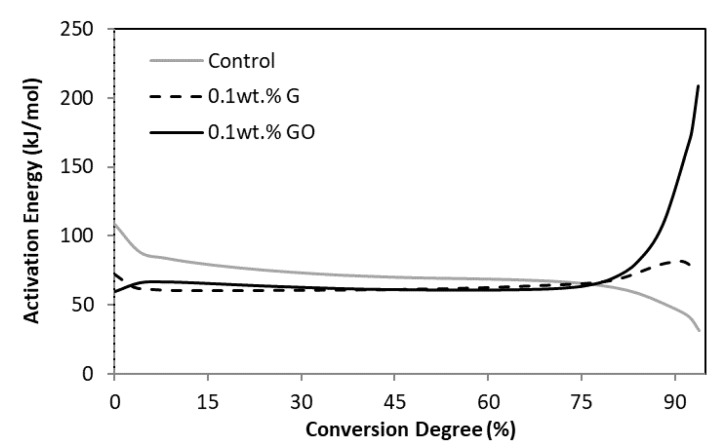
Activation energy vs. conversion degree for the three type of cements.

**Figure 4 materials-12-03146-f004:**
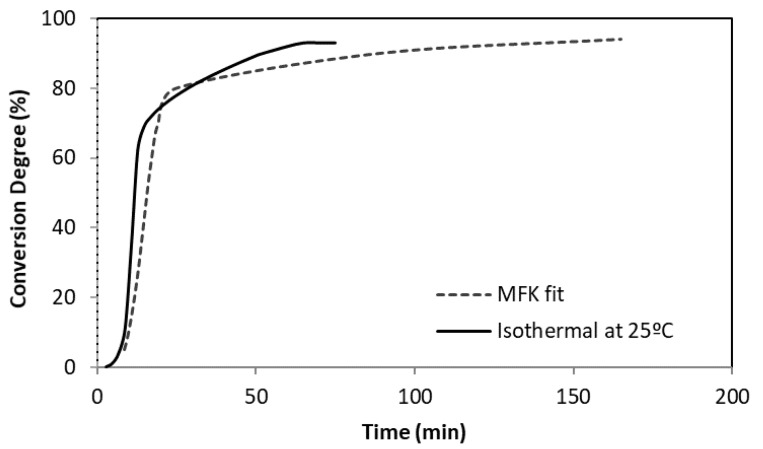
Comparison of the conversion degree between the curing isothermal model and real isothermal process (0.1 wt.% G-PMMA bone cement).

**Figure 5 materials-12-03146-f005:**
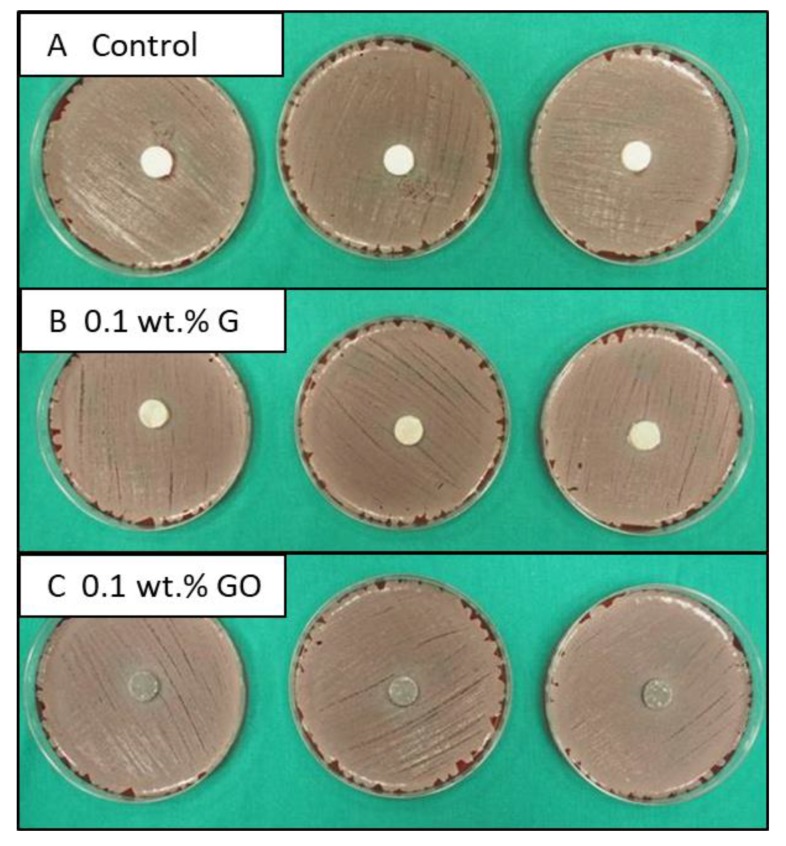
Photographs showing the extent of antimicrobial activity for the (**A**) control, (**B**) 0.1 wt.% G-PMMA bone cement, and (**C**) 0.1 wt.% GO-PMMA bone cement when in contact *Staphylococcus aureus* for an incubation period of 24 h.

**Table 1 materials-12-03146-t001:** Polymerisation enthalpies obtained using the non-isothermal tests for each bone cement type when tested at a different heating rate. The Δ*H_c_* represents the heat released during the non-isothermal polymerisation of the cement and the Δ*H_R_* is the heat associated with the residual monomer. G = graphene; GO = graphene oxide.

Rate (°C/min)	ΔHc (J/mol)			ΔH_R_ (J/mol)		
	5	10	20	5	10	20
Control	81.91	82.80	82.41	4.82	5.32	4.98
0.1 wt.% G	84.49	83.90	84.05	5.01	5.36	5.10
0.1 wt.% GO	84.60	84.49	85.42	5.37	5.42	5.27

**Table 2 materials-12-03146-t002:** Model free kinetics (MFK) simulation data at 25 and 35 °C for the control, 0.1 wt.% GO-, and 0.1 wt.% G-PMMA bone cements.

Properties	Control		G		GO	
Temperature (°C)	25	35	25	35	25	35
Conversion Degree (%)	Time (min)		Time (min)		Time (min)	
5	18.06	5.11	8.56	3.83	10.19	3.11
10	19.65	5.94	9.89	4.49	11.66	4.53
20	20.58	6.75	11.76	5.35	13.64	5.19
30	20.91	7.23	13.23	6.00	14.41	5.66
40	21.03	7.60	14.50	6.55	15.18	6.14
50	21.03	7.90	15.74	7.05	15.95	6.61
60	21.03	8.16	17.17	7.59	17.08	7.09
70	21.03	8.33	19.21	8.30	18.70	7.57
80	21.03	8.50	25.04	10.33	25.49	10.02
90	21.04	9.27	89.26	30.86	659.43	100.12
94	21.04	10.72	164.93	59.01	-	2008.7

**Table 3 materials-12-03146-t003:** Glass transition temperature (Mean ± SD) and the thermal conductivity (Mean ± SD) for the control, GO-, and G-PMMA bone cements.

Cement	Glass Transition Temperature, T_g_			Thermal Conductivity, λ		
	T_g_ (°C)	Difference (%)	*p*-Value	λ (W/m·°C)	Difference (%)	*p*-Value
Control	108.6 ± 0.4			0.176 ± 0.015		
0.1 wt.% G	106.4 ± 1.6	−2.0	0.0852	0.195 ± 0.023	10.9	0.1981
0.1 wt.% GO	108.7 ± 0.2	0.1	0.7049	0.173 ± 0.011	−1.8	0.0963

**Table 4 materials-12-03146-t004:** Viability of the MC3-T3 cells (Mean ± SD) when directly exposed to control, GO-, and G-PMMA bone cements following an incubation period of 72 h.

Cell viability	Control	G	GO
Number of cells (± SD)	6229 ± 556	5752 ± 21	4451 ± 107
Difference vs. control (%)		−7.7	−28.6
*p*-value		0.995	0.454

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
