# Peer review of "Graphene Oxide and Graphene Reinforced PMMA Bone Cements: Evaluation of Thermal Properties and Biocompatibility"

_materials, 2019, doi:10.3390/ma12193146_

Round 1

Reviewer 1 Report

Comments to the authors

I recommend this article can be published with minor revision.

Some comments are below:

Figure 1, the temperature should be provided in x axis. And the second exothermic peaks are not prominent, so these peaks should be highlighted in the inset in figure1. In Table 2, to reach the conversion degree 94%, it required 2008.7min at 35 oC, which is far higher than 90% conversion degree, please check it. Please explain your Tg data (Table 3) more in line 395.

Author Response

Point 1. Figure 1, the temperature should be provided in x axis. And the second exothermic peaks are not prominent, so these peaks should be highlighted in the inset in figure1.

Response 1: The temperature in the x-axis has been included, the second exothermic peak has been indicated in Figure 1 to facilitate the ease of understanding of the data.

Point 2. In Table 2, to reach the conversion degree 94%, it required 2008.7min at 35 ºC, which is far higher than 90% conversion degree, please check it.

Response 2: This data shown in Table 2 corroborates the proposed hypothesis, i.e. the termination step in the case of the GO-PMMA bone cement is delayed. The following paragraph has been introduced after line 376 to comment this interesting aspect:

"It is also interesting to note that for GO powder, the time to progress the polymerisation reaction from 90 to 94% was extremely long (Table 2), i.e. the 90% of conversion was reached in approximately 100 min but it took 31 h to progress the reaction from 90 to 94% of conversion. This observation supports the fact that the termination reaction in the case of the GO-PMMA bone cement was delayed due the presence of the GO powder.”

Point 3. Please explain your Tg data (Table 3) more in line 395.

Response 3:  The following paragraph has been added in line 395:

“DSC analysis from this study indicated a 2% reduction in Tg when 0.1 wt.% G powder was added to the PMMA bone cement. However, this difference was insignificant (p<0.05) and was within the typical standard deviation associated with the DSC technique. Similarly incorporating 0.1 wt.% GO powder into the PMMA bone cement resulted in a 0.09% increases in Tg when compared to the control.”

Reviewer 2 Report

The report is uploaded as attachment. 

Author Response

The statement “Graphene oxide (GO) and graphene (G) have become two of the most interesting nanomaterials 48 due to their unique properties (i.e. high mechanical strength, high specific area, easily 49 functionalisation, high thermal and electrical conductivity, catalyst capacity, etc.) [1–4].” requires some citations on papers that report thermal properties of graphene and GO. The relevant literature are Nature Mater., vol. 10, no. 8, pp. 569–581, 2011; Reports on Progress in Phys., vol. 80, no. 3, p. 36502, Mar. 2017.

Response: The suggested articles have been cited in the manuscript as the references [6] and [7].

Raman spectra or other characterization data confirming that the authors indeed used graphene oxide and graphene in their cements would improve the paper.

Response: The characterization of the used G and GO has been previously published in the following paper. The refence has been included in the manuscript.

“Paz, E.; Ballesteros, Y.; Forriol, F.; Dunne, N. J.; del Real, J. C. Graphene and Graphene Oxide Functionalisation with Silanes for Advanced Dispersion and Reinforcement of PMMA-Based Bone Cements. Mater. Sci. Eng. C 2019, 104, 109946. https://doi.org/10.1016/j.msec.2019.109946.”

Microscopy images of the composites, showing their micro/nanostructure would strengthen the paper.

Response: In a similar maner than in the case of the characterization of the G and Go, the mechanical properties, SEM images of the nanoparticles and SEM Images of the fracture surface of the composites have been previously published in the following paper:

“Paz, E.; Ballesteros, Y.; Forriol, F.; Dunne, N. J.; del Real, J. C. Graphene and Graphene Oxide Functionalisation with Silanes for Advanced Dispersion and Reinforcement of PMMA-Based Bone Cements. Mater. Sci. Eng. C 2019, 104, 109946. https://doi.org/10.1016/j.msec.2019.109946.”

There have been other studies of composites with graphene and GO which reported improvements of various properties. Inclusion of prior works would be proper. Some relevant papers are: S. Ramirez, et al., “Thermal and magnetic properties of nanostructured densified ferrimagnetic composites with graphene - graphite fillers,” Mater. Des., vol. 118, pp. 75–80, 2017; F. Kargar, et al., "Dual-functional graphene composites," Adv. Electron. Mater., vol. 5, no. 1, p. 1800558, 2019.

Response: The suggested articles have been cited in the manuscript as the references [54] and [55].

Reviewer 3 Report

This manuscript describes the effect (or No effect) of incorporation of graphene oxide and graphene powder in PMMA bone cement on thermal properties and biocompatibility. 

This study is original, and the experimental method, data, and results all appear well conceived, executed, and analyzed. Therefore, I recommend this article to be published with minor corrections on some typos or miswordings. Some of them are; in line 49 (easily functionalisation => easy functionalisation), in line 87 (composed by => composed of), in line 96 (can reached => can reach), etc.

Also, the title might be changed more specifically as follows; from "Graphene Oxide and Graphene Reinforced Bone Cements..." to "Graphene Oxide and Graphene Reinforced PMMA Bone Cements..."

Author Response

Point 1. This manuscript describes the effect (or No effect) of incorporation of graphene oxide and graphene powder in PMMA bone cement on thermal properties and biocompatibility.

This study is original, and the experimental method, data, and results all appear well conceived, executed, and analyzed. Therefore, I recommend this article to be published with minor corrections on some typos or miswording. Some of them are; in line 49 (easily functionalisation => easy functionalisation), in line 87 (composed by => composed of), in line 96 (can reached => can reach), etc.

Also, the title might be changed more specifically as follows; from "Graphene Oxide and Graphene Reinforced Bone Cements..." to "Graphene Oxide and Graphene Reinforced PMMA Bone Cements..."

Response 1: The corrections suggested by the Reviewer #2 have been implemented and the manuscript has been carefully read throughout and all typographical errors have been corrected. The title of the manuscript has been also changed as per the Reviewer’s advice.
